# Psychometric Properties for Multidimensional Cognitive Load Scale in an E-Learning Environment

**DOI:** 10.3390/ijerph19105822

**Published:** 2022-05-10

**Authors:** Younyoung Choi, Hyunwoo Lee

**Affiliations:** 1Department of Psychology, Ajou University, Suwon 16499, Korea; 2Biohealth Convergence-Open-Sharing System, Dankook University, Cheonan 31116, Korea; hwlee99@hanyang.ac.kr

**Keywords:** cognitive load scale, e-learning system, validity, reliability, psychometric properties

## Abstract

(1) Background: A learner’s cognitive load in a learning system should be effectively addressed to provide optimal learning processing because the cognitive load explains individual learning differences. However, little empirical research has been conducted into the validation of a cognitive load measurement tool (cognitive load scale, i.e., CLS) suited to online learning systems within higher education. The purpose of this study was to evaluate the psychometric properties of the CLS in an online learning system within higher education through the framework suggested by the Standards for Educational and Psychological Testing. (2) Methods: Data from 800 learners were collected from a cyber-university in South Korea. The age of students ranged from 20 to 64. The CLS was developed, including three components: extraneous cognitive load, intrinsic cognitive load, and germane cognitive load. Then, psychometric properties of the CLS were evaluated including reliability and validity. Evidence relating to content validity, construct validity, and criterion validity were collected. The response pattern of each item was evaluated on the basis of item response theory (IRT). Cronbach’s α was computed for reliability. (3) Results: The CLS presented high internal consistency. A three-factor model with extraneous cognitive load, intrinsic cognitive load, and germane cognitive load was suggested by exploratory and confirmatory factor analysis. This three-factor model is consistent with the previous research into the cognitive load in an offline learning environment. Higher levels of the extraneous cognitive load and intrinsic cognitive load were related to lower levels of academic achievement in an online learning environment, but the germane cognitive load was not significantly positively associated with midterm exam scores, though it was significantly related to the final exam scores. IRT analysis showed that the item-fit statistics for all items were acceptable. Lastly, the measurement invariance was examined through differential item functioning analysis (DIF), with the results suggesting that the items did not contain measurement variance in terms of gender. (4) Conclusions: This validation study of the CLS in an online learning environment within higher education assesses psychometric properties and suggests that the CLS is valid and reliable with a three-factor model. There is a need for an evaluation tool to take into account the cognitive load among learners in online learning system because the characteristics of learners within higher education were varied. This CLS will help instructional/curriculum designers and educational instructors to provide more effective instructions and identify individual learning differences in an online learning environment within higher education.

## 1. Introduction

Cognitive load theory states that optimal learning occurs when the total cognitive load is handled in a way that does not exceed the working memory capacity during cognitive processing [1,2,3,4,5]. Cognitive processing consists of a series of three procedures: (1) selecting appropriate text/imagery for processing the working memory, (2) modeling a mental representation through the selected text and images, and (3) integrating the learner’s prior knowledge with the mental representation [6]. During processing, the degree of knowledge acquisition depends on how efficiently a learner uses the cognitive resources, as the capacity of the working memory is limited [3,4,5], [7]. From this point of view, it is important to understand how the cognitive load can be efficiently managed when learning in instructional design and educational research [8,9].

Optimal learning can occur when learners have adequate cognitive processing capacities in their working memory to handle cognitive load [1,2,3,4,5]. Previous studies report that the cognitive load cannot only explain individual learning differences but also provide the core elements for designing a suitable learning system as a learning system is basically designed to reflect human cognitive systems [2]. The cognitive load consists of three components: extraneous cognitive load, intrinsic cognitive load, and germane cognitive load [10]. The three types of cognitive load are additive, and the total of the three loads combined cannot exceed the working memory resources. Therefore, problem-solving and learning are essentially based on the threshold of the total of the intrinsic, extraneous, and germane cognitive loads. In a study of the instructional design in multimedia learning, reducing the burden of the extraneous cognitive load and stimulating the germane cognitive load helped learners to form study-center knowledge constructions [6].

The cognitive load should be effectively managed for optimal learning in a learning system [11,12,13]. The three components of the cognitive load are associated with learning outcomes, the results of a knowledge-based test, and task performance [12]. Therefore, measuring a learner’s cognitive load in a learning system can offer useful information to facilitate learning by efficiently managing and controlling the cognitive load [11,12,13]. In multimedia learning environments, measuring the cognitive load is also used to identify individual learning differences as well as to assess whether a multimedia learning system is properly constructed for learning [6]. Accordingly, research has proposed various tools for measuring the cognitive load in a learning environment (e.g., [14,15]).

### 1.1. Cognitive Load Theory

The cognitive load is defined as the total amount of cognitive resources imposed on the learner’s cognitive system when performing a task [10]. Cognitive load theory proposes that knowledge, skill, and ability acquisitions depend on how efficiently a learner manages cognitive load, and so an instruction should be designed to reduce cognitive load [3,4,5,7]. Accordingly, cognitive load theory can explain the design elements needed for efficient learning environments including enhancing knowledge acquisition and facilitating learners’ sense of belonging, as well as to identify individual learning differences.

There are three main types of cognitive load: extraneous load, intrinsic load, and germane cognitive load, according to cognitive load theory [4,9,16]. These cognitive load components should be managed to optimize a learning effect [1,2,3,4,5]. Specifically, intrinsic cognitive load is a factor directly related to the task diversity and task complexity that a learner should handle. Intrinsic cognitive load is dependent on a learner’s previous learning experiences and pre-requisite learning. The higher the difficulty and complexity of a task or problem, the more information there is to be processed and the higher intrinsic cognitive load that is required [17,18,19].

Extraneous cognitive load is fed by inappropriate instructional design and methods that increase the burden on learners’ working memory. In this way, elements contributing to extraneous cognitive load result in a decline of learning effect as extraneous cognitive load is a hindrance to cognitive processing [8,20]. Appropriate instructional interventions should be designed so that extraneous cognitive load is minimized [14,21]. It is not simple to construct an e-learning system within higher education due to the various characteristics of leaners [22,23]. When intrinsic cognitive load is optimal and extraneous cognitive load is low, learners can engage in their learning environments with more sense of belongingness [24].

Finally, germane cognitive load arises when a learner makes a cognitive effort to transfer information in the working memory to the long-term memory. Germane cognitive load occurs when new information is integrated with the pre-existing inherent knowledge. Therefore, germane cognitive load is positively associated with learning achievement through higher motivation and engagement to learning [4,10]. For instance, germane cognitive load can be activated by the self-explanation principle [25,26]. Additionally, the variability of practices when performing a task facilitates the increase in germane cognitive load [27,28]. Moreover, the contribution and emotional engagement on a course can be a significant factor that effects germane cognitive load.

Given the characteristics of these cognitive loads, effective instructional interventions are designed to minimize extraneous cognitive load and intrinsic cognitive load arising from the working memory while maximizing germane cognitive load. Accordingly, instructional design should greatly consider cognitive load components. Proper measurement tool in terms of the different types of cognitive load would help instructional designers and educational researchers to better understand effective and efficient instructional design. Moreover, the learners possessing different levels of intrinsic and germane cognitive loads can be considered in designing instructions. Therefore, measuring cognitive load has been a critical issue in the areas of instructional design [14,15].

### 1.2. Measuring Cognitive Load

A tool for measuring cognitive load helps to efficiently manage and control cognitive loads [11,12,13]. Many studies have been conducted attempting to develop a scale to measure cognitive load (e.g., [14,15,29,30]). Studies report that measuring cognitive load enables instructional designers and educators to efficiently manage and control cognitive loads [11,12,13]. For example, Cierniak, Scheiter, and Gerjets [31] proposed a cognitive load scale with three items: extraneous cognitive load (i.e., “How difficult was it to learn with the material?”), intrinsic cognitive load (i.e., “How easy or difficult do you consider theory at this moment?”), and germane cognitive load (i.e., “How much did you concentrate during learning?”). Leppink et al. [10] suggested a further cognitive load scale with the same three factors. However, in this study, germane cognitive load was not essentially distinguished from intrinsic and extraneous cognitive loads. Furthermore, germane cognitive load was not statistically significantly associated with a learner’s task performance [10]. Similarly, Jiang and Kalyuga [32] proposed a two-factor model for measuring cognitive load with intrinsic and extraneous cognitive loads rather than considering germane cognitive load as an independent cognitive processing factor in the cognitive load. These findings contrast with the traditional cognitive load theory with three independent factors, i.e., intrinsic, extraneous, and germane cognitive loads [16,24,33,34].

Validation studies relating to a cognitive load measurement tool have mostly been conducted in offline learning environments [33,34,35,36]. In addition, the measurement tools have been applied without distinction between online and offline learning environments. Yet, there are many learning elements that differ between online and offline learning environments [37]. For example, learning processing relies on instructional design in a multimedia environment more than in a face-to-face classroom learning environment [37,38,39]. Moreover, self-regulated learning ability is more required to improve the academic performance in online learning than offline learning environments [40]. The principles of instructional design for multimedia learning focus on constructing a learning system by reducing the burdens of extraneous cognitive load on learners and promoting germane cognitive load [6]. Nevertheless, only a few attempts have been made to develop instruments to measure these different types of cognitive load that are suitable for an e-learning environment. In addition, research rarely assesses the psychometric properties (e.g., validity and reliability) of the CLS in an e-learning system. Andersen and Makransky (2021) studied the validation of developing a multidimensional cognitive load scale for the virtual environment. They additionally divided extraneous load into three concepts including noises in the environment and distractions from devices, instruction, and explanations [41].

Recently, the educational environment in face-to-face learning has gradually expanded to encompass an online educational environment [23]. A learning environment using online multimedia has been especially emphasized in the post-coronavirus era [42]. It has been of high interest to develop a cognitive load measurement tool that is suited to online learning environments. A more applicable measurement of the cognitive load in an online learning environment could help us to understand how online instructional formats and learner characteristics may function differently during e-learning activities. The CLS may facilitate various empirical studies on how to design more effective online instructions and understand the learning differences among learners in an e-learning environment.

### 1.3. The Present Study

The main purpose of this study was to develop a cognitive load scale for an e-learning environment. To that end, we constructed the CLS on the basis of cognitive load theory, which consists of three conceptual factors: extraneous load, intrinsic load, and germane cognitive load. Then, we evaluated the psychometric properties of the CLS, including its reliability and validity, following the framework suggested by the Standards for Educational and Psychological Testing [43]. We first computed the content validity index (CVI) to collect evidence related to content validity. We also analyzed the internal structure of the CLS for construct validity by conducting exploratory and confirmatory factor analysis (EFA and CFA). In addition, we examined the criterion validity by computing the correlation values with academic achievement using the midterm and final grades. Previous studies reported that the extraneous and intrinsic loads are negatively related to a learner’s achievement [10] and that the germane cognitive load is positively associated with achievement, though not significantly [10]. IRT is able to provide information regarding each item’s measurement errors across different levels of cognitive load. In this study, we also examined each item’s performance by using item response theory (IRT [44,45]) to collect evidence of each item’s response pattern across different levels of cognitive load. IRT was also used to evaluate measurement invariance using DIF analysis. Lastly, the reliability was evaluated by Cronbach’s alpha.

## 2. Methods

### 2.1. Participants

The study was conducted using data collected from 800 students in a cyber-university. The class sampled was an online course for introductory statistics to social science. All students took a midterm and final exam during the course. When a student enters the lecture, the student can see the survey window. Moreover, if the student agrees to the survey, all survey statements are shown. It is not necessary for all participants to answer every item. The items are all nine questions with a five-point Likert scale ranging from 1 (not at all) to 5 (to a very large extent). The items were administered in Korean. CLS was translated and verified by the experts related to this area. The participants provided informed consent. The study was given IRB exemption from the institution’s ethical review committee because the survey did not contain any sensitive and personal identifiable information. Table 1 shows the descriptive statistics of the subjects.

### 2.2. Data Analysis

Descriptive statistics of all the measures were computed according to the mean, standard deviation, skewness, and kurtosis. For the content validity, CVI (content validity index) was computed on the basis of the opinions of subject experts. For the construct validity, exploratory and confirmatory factor analysis were conducted using Mplus version 8 [46]. The exploratory and confirmatory factor analysis were conducted and randomly split into two sets of sample data. Oblique (geomin) rotation was used because all three sub-factors were theoretically correlated with each other. Lastly, the measurement invariances of all items were evaluated using the differential item functioning method (DIF [47]) using IRTpro [48]. For the reliability, Cronbach’s alpha of each sub-factor was computed using SPSS version 24.0.

## 3. Results

### 3.1. Descriptive Statistics

The descriptive statistics of three factors were computed according to the mean, standard deviation, skewness, kurtosis, and correlation (Table 2).

### 3.2. Validity Evidence

#### 3.2.1. Content Validity Index

The CVI (content validity index [49,50]) was computed for the content validity of the CLS. The average CVI value across all items was 0.86. This means that most content experts in this study agreed that the sub-factors’ operational definitions and the items were generally acceptable.

#### 3.2.2. Internal Structure (Factor Structure)

Exploratory and confirmatory factor analysis were conducted with two different datasets to analyze the construct validity of the CLS. The Kaiser–Meyer–Olkin (KMO) measure of sampling adequacy (0.66) and Bartlett’s sphericity test (*p* < 0.0001) were reported. In exploratory factor analysis (EFA), the three-factor solution was the most reasonable and interpretable, accounting for 50.89% of the variance. The factor loadings of each item from the EFA are shown Table 3 and Figure 1. In the CFA, the final model-fit statistics with the three-factor model had an appropriate fit, although the chi-square was relatively large due to the sample size. The CFI was 0.965, TLI was 0.96, RMSEA was 0.071, and SRMR was 0.035 (Table 4) according to the criterion, i.e., CFI and TLI were above 0.90, RSMSA was below 0.08, and SRMR was below 0.10 [51,52,53,54].

#### 3.2.3. Evidence Based on Relation with Other Variables (Criterion Validity)

The criterion validity was evaluated using the correlation analysis with a learner’s midterm and final exam scores (Table 5). The extraneous and intrinsic cognitive loads were negatively associated with the midterm and final exam scores, while the germane cognitive load showed a moderate positive correlation with the midterm and final exam scores.

#### 3.2.4. Evidence Based on Response Patterns

The probability response functions (PRFs) for each item in CLS were drawn (Figure 2). The PRFs of all items in CLS suggest that all categories of each item contained measurement information.

#### 3.2.5. Measurement Invariance

The measurement invariance in terms of gender was evaluated for each item on the basis of DIF analysis. No items showed measurement variance in terms of gender. This meant that the difficulty and discrimination levels of all items were not considered to differ by gender. There was no difference between the men and women for any of the items (Table 6).

### 3.3. Item Characteristics and Reliability

Cronbach’s alpha of the final CLS was 0.74, and for each factor was 0.72 for the extraneous cognitive load, 0.74 for the intrinsic cognitive load, and 0.70 for the germane cognitive load. Moreover, we conducted the McDonald’s Omega coefficient (0.76 for extraneous cognitive load, 0.78 for intrinsic cognitive load, and 0.73 for germane cognitive load).

## 4. Discussion

The aim of this study was to validate the CLS for use with online college students. For that purpose, we collected validity evidence based on content, factor structure, and other related measures such as academic achievement. Moreover, we evaluated the response pattern of each item and measurement invariance in terms of gender. In addition, we analyzed the reliability of the CLS. A statistics online course was chosen for this study as the previous studies have been conducted in the statistics knowledge domain [31,33,55].

Overall, this study found that the CLS possesses good psychometric properties. The three-factor model was suggested, which is consistent with previous research (e.g., [56]). The CLS showed good content validity according to the evaluation by subject experts in e-learning within higher education. Specifically, each sub-domain with intrinsic cognitive load, extraneous cognitive load, and germane cognitive load was appropriately applicable to the online learning environment within higher education, and almost every item appropriately reflected the subscale to which it belonged. On the basis of content analysis, it was confirmed that the CLS was developed to be more suitable to an online learning system.

In this study, we conducted EFA and CFA, finding that the CLS has a stable three-factor structure. These findings are consistent with previous research (e.g., [10]). This result implies that three factors—i.e., extraneous cognitive load, intrinsic cognitive load, and germane cognitive load—are suited to online higher educational institutes for measuring a learner’s cognitive load. In studying the intrinsic cognitive load, we utilized three items: (1) the topics covered in this lecture were very difficult in terms of my previous knowledge, skills, and educational experiences; (2) the concepts and definitions in this lecture were complex in terms of my previous knowledge, skills, and educational experiences; (3) the class objectives, quizzes, and class activities with other leaners were difficult in terms of my previous knowledge, skills, and educational experiences. As in previous studies, the three items for studying the intrinsic cognitive load were directly related to the class difficulty and complexity and learning activities with others that a learner should undertake. Accordingly, if a learner thinks a course involves a higher level/complexity of learning activities, knowledge, skills, and abilities, a higher intrinsic cognitive load is reported [17]. An additional concept was not considered in terms of the online learning system. For the extraneous cognitive load, we developed three items to measure the learner’s perceptions of appropriate instructional design and methods [14,21]: (1) the format of the lecture screen for this lecture is designed to be easy to learn; (2) the functions for learning activities in this e-learning course (e.g., buttons and menus for question and answer session, discussion session with other learners, learning activities with other learners, quizzes, exams) are conveniently provided; (3) the instruction is designed for supporting adaptation to learning environment and the sense of belongness to the course. Since many different types of devices can be used in online learning system in term of contents, device characteristics need to be investigated in future study. Finally, we developed three questions to measure the germane cognitive load: (1) How much did you concentrate and be engaged during the lecture? (2) How much did you put in in terms of mental and emotional effort and time for this class? (3) Did this course enhance the motivation of learning new knowledge, understanding, and application of skills in the domain? These items for measuring the germane cognitive load were related to the cognitive effort required to master knowledge, skills, and abilities and integrate the new information with the pre-existing inherent knowledge [28].

As expected from theories and previous empirical studies [12], learners within e-learning higher education with higher levels of extraneous and intrinsic cognitive loads showed lower levels of academic achievement as measured by their midterm and final exam scores in this online course. There is no significant association between higher levels of germane cognitive load and midterm exam scores, although a significant relationship was found with the final exam scores. These results are consistent with previous research showing negative associations between the cognitive load—related to the extraneous and intrinsic cognitive loads—and indicators of academic achievement and knowledge acquisition. These findings support the notion that the CLS is a valid instrument with which to measure learners’ cognitive load in an online learning system.

On the basis of the results of response pattern analyses for each item using IRT, all the items were appropriate for measuring the construct of cognitive load. Specifically, each item showed that the intervals between adjacent categories in the five of the Likert scales provide meaningful information. In addition, the results for the measurement invariance of each item based on DIF analysis in terms of gender indicated that no gender-related measurement variance was detected. Lastly, the CLS showed good internal consistency, supporting the notion that the instrument could serve as a reliable tool for assessing the CL in an online learning system. Therefore, this study provides evidence of the validity and reliability of the CLS in an online learning system within higher education.

This study has several limitations that should be taken into consideration. First, although the purpose of this study was to develop a self-report scale for an online learning higher education system, many different methods for measuring the cognitive load have been reported, such as physiological measures using pupil size dilation [57], heart rate [36], electroencephalograms [58], and brain activation based on fMRI [59,60]. Future research should include relational analysis between the self-reporting CLS and other measures for collecting validity evidence. Second, the current study only focused on adult learners using an e-learning system. Future studies need to examine whether the instrument shows validity for different ages including children and adolescents since e-learning systems have recently become more popular across the educational spectrum. Third, this study only examined the measurement invariance in terms of gender. Future studies of measurement invariance between online and offline learning systems will provide more useful information to understand the cognitive load in e-learning systems. Furthermore, a comparison study of the CLS between online and offline learning systems will be helpful for discovering the unique effects of the cognitive load in an e-learning system. Finally, future validation studies in e-learning courses in different domains will provide validity evidence about the stability of the three-factor model in the CLS.

Nevertheless, considering how the educational environment in face-to-face learning has gradually expanded to encompass an online educational environment, a CLS that is more suited to e-learning systems can be a useful measurement tool. The effective measurement of the cognitive load in e-learning within higher education would help instructional designers and instructors to better understand why learning outcomes may differ in terms of e-learning instructional formats or various characteristics among learners.

This study provides validity and reliability evidence of the CLS in an online learning system. The findings from the CLS contribute to developing more effective e-learning instructions, thereby reducing the burdens of the extraneous and intrinsic cognitive loads on learners and promoting the germane cognitive load. The CLS can help us to obtain a better understanding of instructional effects on learners with different levels of learner expertise from the intrinsic and germane cognitive loads. In addition, the CLS can identify leaners’ motivation, engagement, and ability to learn and achieve in a course. Therefore, the current study is meaningful for researchers, educators, and practitioners, helping them to better understand the cognitive loads of adult learners by identifying the effectiveness and efficiency of e-learning environments as a function of different instructional formats and learner characteristics.

## Figures and Tables

**Figure 1 ijerph-19-05822-f001:**
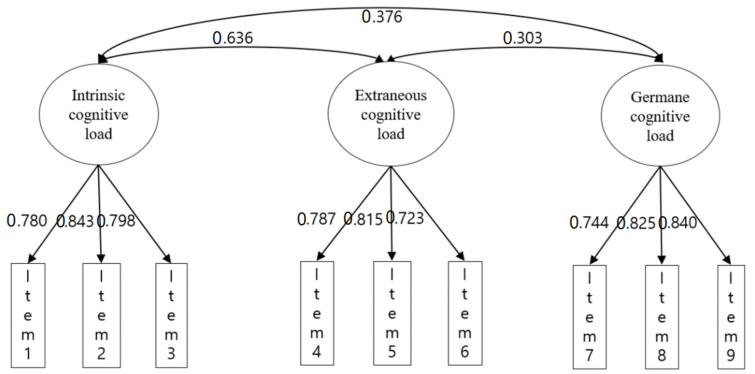
Internal structure of CLS.

**Figure 2 ijerph-19-05822-f002:**
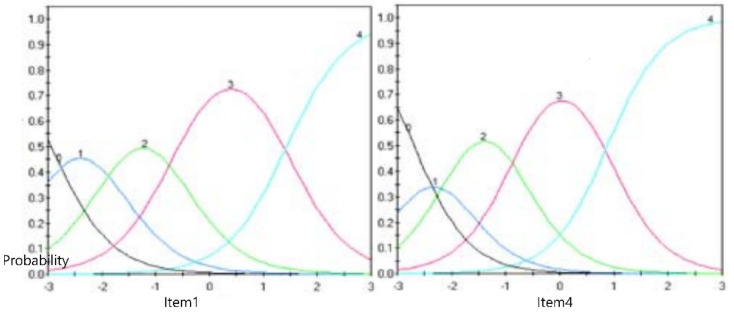
Exemplary probability response function curves (items 1 and 4).

**Table 1 ijerph-19-05822-t001:** Descriptive statistics of subjects.

		Percent	Count
Educational level	Freshman	45.60%	365
	Sophomore	24.50%	196
	Junior	14.10%	113
	Senior	15.80%	126
Age	20	41.80%	334
	30	27.10%	217
	40	15.90%	127
	50	15.30%	122
Gender	Male	65.80%	526
	Female	34.30%	275
Job status	Full time	51.90%	415
	Part time	30.80%	246
	No	17.40%	139

**Table 2 ijerph-19-05822-t002:** Descriptive statistics of three cognitive load components.

Sub-Factor	Intrinsic Cognitive Load	Extraneous Cognitive Load	Germane Cognitive Load
Intrinsic cognitive load	1	0.420 **	0.2120 **
Extraneous cognitive load		1	0.3130 **
Germane cognitive load			1
	**Mean (SD)**	**Skewness**	**Kurtosis**
Intrinsic cognitive load	9.77 (3.70)	0.391	1.20
Extraneous cognitive load	8.77 (2.24)	0.065	0.42
Germane cognitive load	8.82 (2.40)	0.081	0.49
		Mean	SD
Topics covered in this lecture were very difficult in terms of my previous knowledge, skills, and educational experiences		3.20	1.25
Concepts and definitions in this lecture were complex in terms of my previous knowledge, skills, and educational experiences		3.42	1.02
Class objectives, quizzes, and class activities with other leaners were difficult in terms of my previous knowledge, skills, and educational experiences		3.15	1.50
Format of the lecture screen for this lecture is designed to be easy to learn		3.01	1.12
The functions for learning activities in this e-learning course (e.g., buttons and menus for question-and-answer session, discussion session with other learners, learning activities with other learners, quizzes, exams) are conveniently provided		3.21	0.24
The instruction is designed for supporting adaptation to the learning environment and the sense of belongness to the course; finally, we developed three questions to measure the germane cognitive load		2.55	0.88
How much did you concentrate and be engaged during the lecture?		2.98	0.62
How much did you put in in terms of mental and emotional effort and time for this class?		2.68	0.89
Did this course enhance the motivation of learning new knowledge, understanding, and application of skills in the domain?		3.16	1.31

** *p* < 0.01.

**Table 3 ijerph-19-05822-t003:** Factor loading of EFA and unstandardized and standardized coefficients of CFA.

Items	EFA			CFA	
	Intrinsic Cognitive Load	Extraneous Cognitive Load	Germane Cognitive Load	Standardized Coefficient	S.E.
Item 1	**0.721**	−0.124	−0.047	0.770	0.019
Item 2	**0.824**	−0.059	−0.061	0.831	0.015
Item 3	**0.812**	−0.077	−0.044	0.776	0.017
Item 4	0.145	**0.649**	0.053	0.772	0.018
Item 5	0.124	**0.627**	0.114	0.802	0.019
Item 6	−0.002	**0.554**	−0.272	0.713	0.025
Item 7	0.083	0.001	**0.693**	0.734	0.021
Item 8	0.044	−0.186	**0.551**	0.816	0.023
Item 9	0.027	0.123	**0.678**	0.831	0.024

Note: the order of all items is the same as Table 2. The bold numbers indicate relatively higher than others.

**Table 4 ijerph-19-05822-t004:** Model-fit statistics of CFA.

Model	x^2^ (df)	TLI	CFI	RMSEA [90 Percent C.I.]	SRMR
Three factors	74.23 (24)	0.965	0.960	0.071 (0.045, 0.078)	0.035

**Table 5 ijerph-19-05822-t005:** Correlation with midterm and final exam scores.

Sub-Factor	Extraneous Cognitive Load	Intrinsic Cognitive Load	Germane Cognitive Load
Midterm exam	−0.382 **	−0.351 **	0.023
Final exam	−0.314 **	−0.415 **	0.213 **

** *p* < 0.01.

**Table 6 ijerph-19-05822-t006:** Results of DIF analysis.

Item	$x^{2}$	*p*
Item 1	1.10	0.97
Item 2	3.00	0.41
Item 3	2.30	0.74
Item 4	4.10	0.48
Item 5	3.50	0.59
Item 8	1.90	0.86
Item 9	5.00	0.43

Note: the order of all items is the same as Table 2.

## Data Availability

Data has been collected during the study and can be requested to the corresponding author.

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
