# Peer review of "Psychometric Properties for Multidimensional Cognitive Load Scale in an E-Learning Environment"

_ijerph, 2022, doi:10.3390/ijerph19105822_

Round 1
Reviewer 1 Report
The authors attempted to develop the Cognitive Load Scale (CLS) in E-Learning Environment based on 800 students from Cyber University in South Korea. The study was well implemented, there are some minor issues need to be addressed by the authors:
1) Please further conceptualise the three main types of cognitive load (p. 2-3), i.e. extraneous load, intrinsic load, and germane cognitive load and its relationship with other construal-related measures, such as academic performance and other measures. It will be easier for the reader to further understand the criterion validity of the scale.
2) What are the problems of the existing cognitive load measurement tools and the urgency for developing a new scale (p. 3)?
3) To avoid the issue of overfitting, please avoid performing EFA and CFA on the same dataset (Fokkema & Greiff, 2017).
4) Why the authors used both CTT and IRT to develop the scale? Please provide more justifications (Jebb, Ng, & Tay, 2021).
5) In addition to the Cronbach’s alpha, please also report the McDonald’s Omega coefficient (Beland, Cousineau, & Loye, 2017; Gignac, Reynolds, & Kovacs, 2019).
6) In the methods section, please report whether the participants provide the informed consent and the study was endorsed by the institution’s ethical review committee.
7) Please report the descriptive statistics of all the items (including question/item of the scale).
8) The df/chi-square value on Table 4 is over 3, which means the model maybe problematic (Kline, 2005). Please address this issue.
9) Typos: “leaners” (p. 2), “RSMSE” (p. 6). Please revise it.
References
Beland, S., Cousineau, D., & Loye, N. (2017). Using the Mcdonald's Omega Coefficient instead of Cronbach's Alpha. Mcgill Journal of Education, 52(3), 791-804. Retrieved from <Go to ISI>://WOS:000437472500013
Fokkema, M., & Greiff, S. (2017). How Performing PCA and CFA on the Same Data Equals Trouble Overfitting in the Assessment of Internal Structure and Some Editorial Thoughts on It. European Journal of Psychological Assessment, 33(6), 399-402. doi:10.1027/1015-5759/a000460
Gignac, G. E., Reynolds, M. R., & Kovacs, K. (2019). Digit Span Subscale Scores May Be Insufficiently Reliable for Clinical Interpretation: Distinguishing Between Stratified Coefficient Alpha and Omega Hierarchical. Assessment, 26(8), 1554-1563. doi:10.1177/1073191117748396
Jebb, A. T., Ng, V., & Tay, L. (2021). A Review of Key Likert Scale Development Advances: 1995–2019. Frontiers in Psychology, 12(1590). doi:10.3389/fpsyg.2021.637547
Kline, R. B. (2005). Principles and practice of structural equation modeling (2 ed.): New York: Guilford Press.
Author Response
1) Please further conceptualise the three main types of cognitive load (p. 2-3), i.e. extraneous load, intrinsic load, and germane cognitive load and its relationship with other construal-related measures, such as academic performance and other measures. It will be easier for the reader to further understand the criterion validity of the scale.
We added the previous researches related to this. For instance, Mayer et al (2005) and Jong (2010) suggested that the components of the cognitive load itself and their interactivities among the components effects academic performances, task performances, and the results of knowledge based tests.
2) What are the problems of the existing cognitive load measurement tools and the urgency for developing a new scale (p. 3)?
The validation of cognitive load scale has been mostly conducted in offline learning environment. Therefore, the previous studies of cognitive load scales have not fully investigated the leaners’ cognitive load characteristics in online learning environment. We addressed this issue on the page 3.
3) To avoid the issue of overfitting, please avoid performing EFA and CFA on the same dataset (Fokkema & Greiff, 2017).
As your recommended, we performed the EFA and CFA with different data sets using 400 samples. To do this, we randomly spitted data into two data samples.
4) Why the authors used both CTT and IRT to develop the scale? Please provide more justifications (Jebb, Ng, & Tay, 2021).
The CTT mostly focuses on the assessment/survey quality instead of item-level quality. Also, we would like to explain the measurement errors in term of different levels of construct (i.e., cognitive load). In addition, we would like to evaluate the measurement invariance of each item. We addressed the usefulness of IRT method addition to CTT.
5) In addition to the Cronbach’s alpha, please also report the McDonald’s Omega coefficient (Beland, Cousineau, & Loye, 2017; Gignac, Reynolds, & Kovacs, 2019).
We computed the McDonald’s Mega coefficient and added the values in the reliability section.
6) In the methods section, please report whether the participants provide the informed consent and the study was endorsed by the institution’s ethical review committee.
We reported the participants’ informed consent and the study was given by IRB exemption from the institution’s ethical review committee because the survey didn’t contain any sensitive and personally identifiable information
7) Please report the descriptive statistics of all the items (including question/item of the scale).
We addressed all items’ questions and scales in the result, discussion and method sections.
8) The df/chi-square value on Table 4 is over 3, which means the model maybe problematic (Kline, 2005). Please address this issue.
Based on the Kline (2005), sample size over 400 would lead the large chi-square value. So that other model fit statistics should be considered to evaluate the model. We added this issue the result section.
9) Typos: “leaners” (p. 2), “RSMSE” (p. 6). Please revise it.
We corrected them
Reviewer 2 Report
I did not find where this was performed:
In order for the contribution to be extended to other contexts, I suggest adding in the methodology section a description of the instrument with the dimensions and items used for the evaluation.
Author Response
Dear the Reviewer,
Thank you so much for your time. We modified our paper. Regarding the items amd dimension information, we provided the information in the discussion section using a description instead of a table. You are able to see the information of all items in terms of three dimensions in the discussion section.
Thank you
Sincerely,
This manuscript is a resubmission of an earlier submission. The following is a list of the peer review reports and author responses from that submission.
Round 1
Reviewer 1 Report
Assessing Cognitive Load in E-Learning Environment: Psychometric Properties of Cognitive Load Scale (CLS) for E- Learning
Summary of the Manuscript
The focus of this manuscript was to develop a psychometric scale to measure cognitive loads for e-learning.
Overall, this manuscript’s writing and readability was sound quality, and the ideology is unique; however, the manuscript needs to go through some reviews. Moreover, doing some additional analyses will enhance the analytical quality of the manuscript. Some minor changes need to be made throughout the manuscript.
Manuscript Comments:
- Item pool: Did you use 9 items of CLS? Moreover, you showed 9 items in CFA model, but later you used 12 items for DIF analysis. Could you please include explanation for the additional 3 items?
- You might be interested in the finding of “Andersen, M. S., & Makransky, G. (2021). The validation and further development of a multidimensional cognitive load scale for virtual environments. Journal of Computer Assisted Learning, 37(1), 183-196.”
- I would also like to suggest authors to incorporate how each item was measured in the survey i.e., measurement scale. It will make the manuscript more convenient for the readers.
- Page 2[ 94]: Please use similar terms for extrinsic cognitive load, i.e., either extraneous or extrinsic.
- Page 2[96]: Please check for spelling mistake.
- Page 3[100-101]: You may consider rephrasing the sentence.
- Page 3[103]: Probable spelling mistake. Did you mean “Through”?
- Page 3[104]: Please indicate the direction of the facilitation i.e., does it reduce the cognitive load or increase the load? Also, elaborate a little on this issue.
- Page 3[108]: The concept is good, but the sentence is not clear. I would recommend you break down the sentence.
- Page 3[114]: Did you mean “researchers”? Also, did you mean “learners with different intrinsic and germane cognitive load capacities?”. Please rephrase the sentence.
- Page 3[142]: I would suggest authors to rephrase the sentence.
- Page 3[149]: I would rather suggest saying “Rarely” instead of “No researches”.
- Page 3[150]: I would suggest authors to rephrase the sentence.
- Page 4[158]: I would suggest authors to check for grammatical error.
- Page 4[179]: Sample size 600? Did you collect data from exactly 800 students? Or did you drop any responses? Also, was there any missing value in the data? Please mention that as well. Moreover, what is the response rate? Did you send the survey to 800 students or more? Also, please report when did you do the survey? Survey period. Timing of the survey, during class period or at respondent’s leisure time? Were they allowed to skip the survey?
- Page 4[183]: Could you please explain the education level in terms of some international standard i.e., what does 1/2-year means?
- Page 5[ 186]: I would suggest authors to consider rephrasing the sentence.
- Page 5[205]: I would like to recommend authors to incorporate measures for Composite reliability (CR), Average variance extracted (AVE) for each construct, and discriminant validity for the CFA model. The current measures are good, but this will enhance the scale properties.
- Page 6 [225]: In the section “Evidence based relation with other variables” section, I would like recommend authors to explain the correlation more appropriately. The correlation comments, and the correlation values do not support each other. Moreover, what values or measures of extraneous, intrinsic, and germane cognitive load were used to calculate the correlations with midterm or final exam scores?
- Page 8[304]: I would suggest authors to consider rephrasing the sentence.
Reviewer 2 Report
The authors attempted to validate the Cognitive Load Scale (CLS) based on a sample recruited from an online university. There are several major issues need to be addressed by the authors:
- The title is a bit redundant, please consider to revise it.
- The authors should further elaborate/discuss the controversy related to the different CLS versions reported in the literature. The CFA results should also evaluate those versions reported in the literature to compare which one with better construct validity.
- There is a potential problem of overfitting by running EFA and CFA on the same data set. The authors may consider to remove EFA and report the McDonald’s Omega value.
- Did the study approved by any research ethics committee within the university? Did the participants provided informed consent?
- In Table 2, please also provide the results such as mean scores of each items, corrected item-total correlations, and Cronbach’s alpha, if item deleted.
- The scale was administered in English? If not, please report the translated procedure.
- The Figure 1 results are confusing, the authors only show item 1, 2 and 3. How about items 4-9?
- The CFA results failed to fulfill the requirement of adequate model fit, as the chi-square/df > 3.0. The proposed three factor structure maybe problematic.
- The results reported in Table 5 is out of the place. It seems the authors try to evaluate the concurrent validity of the scale. But it was not discussed in the methods section, such as how to measure the results. Was there any ethical issues involved when extracting the exam marks from the participants. Did the existing CLS literature also used the exam results to evaluate the concurrent validity? Please provide the justifications from the literature.
- Why the authors use IRT together with the traditional scale validation methods? More justifications are required in the methods section.
Reviewer 3 Report
The study is well-supported and relevant to the current environment.
In order for the contribution to be extended to other contexts, I suggest adding in the methodology section a description of the instrument with the dimensions and items used for the evaluation.